# Demographic and Behavioral Risk Factors Predict Functional Limitations Associated with Subjective Cognitive Decline in Americans with Chronic Obstructive Pulmonary Disease: A Secondary Analysis

**DOI:** 10.3390/ijerph21030278

**Published:** 2024-02-28

**Authors:** Michael Stellefson, Min Qi Wang, Sarah Flora, Olivia Campbell

**Affiliations:** 1Department of Health Science, The University of Alabama, Tuscaloosa, AL 35487, USA; sflora@crimson.ua.edu (S.F.); okcampbell@crimson.ua.edu (O.C.); 2Department of Behavioral and Community Health, University of Maryland, College Park, MD 20742, USA; mqw@umd.edu

**Keywords:** COPD, BRFSS, subjective cognitive decline, functional limitations

## Abstract

Prior research indicates that subjective cognitive decline (SCD) affects approximately one-third of older adults with Chronic Obstructive Pulmonary Disease (COPD). However, there is limited population-based research on risk factors associated with SCD-related functional limitations within this vulnerable subgroup. A secondary data analysis of 2019 Behavioral Risk Factor Surveillance System data was conducted to address this gap, focusing on Americans ≥45 years old with COPD (N = 107,204). Several sociodemographic and health-related factors were independently associated with SCD-related functional limitations. Retired and unemployed individuals were significantly more likely to require assistance with day-to-day activities due to memory loss or confusion compared to employed individuals (AOR = 3.0, 95% CI: 1.2–8.0; AOR = 5.8, 95% CI: 3.01–1.5, respectively). Additionally, unemployed individuals were over five times more likely to report confusion or memory loss affecting social activities (AOR = 5.7, 95% CI: 2.9–11.0). Disparities were also observed among different racial groups, with Black/African Americans (AOR = 4.9, 95% CI: 2.3–10.4) and Hispanics (AOR = 2.4, 95% CI: 1.2–4.7) more likely than White and non-Hispanic people, respectively, to give up daily chores due to SCD. Our findings underscore the need for culturally sensitive interventions to address functional limitations faced by retired, unemployed, and minority adults with COPD and SCD.

## 1. Introduction

Chronic Obstructive Pulmonary Disease (COPD) is a persistent health condition characterized by airflow limitations, making it the sixth leading cause of death in the United States (US) [1]. The burden of COPD is expected to increase significantly in the coming years, with an estimated 4.5 million people projected to die annually due to COPD and related conditions by 2030, and this number is expected to rise to 5.4 million by 2060 [2]. The impact of COPD on individuals is profound, as it causes debilitating lung-related symptoms such as chest tightness, breathing issues, and shortness of breath, which worsen over time and can lead to difficulties in performing activities of daily living (ADLs) [1]. Furthermore, COPD is associated with weaker muscle strength in every muscle group in both the upper and lower extremities, which can contribute to declines in mobility and independence [3]. As a result, individuals with moderate to severe COPD experience challenges in performing basic ADLs such as walking, putting on shoes, and taking a shower, leading to reduced engagement in hobbies and increased social isolation compared to their non-COPD counterparts [4,5]. 

The burden of COPD is further exacerbated by the presence of comorbidities, with the majority of individuals with COPD living with several other health conditions [6,7]. This burden is exceptionally high in individuals with cognitive impairment, a common comorbidity experienced by people with COPD, which leads to mild to severe memory loss and confusion that significantly impacts self-management and quality of life [8,9,10]. Subjective cognitive decline (SCD), characterized by self-reported experiences of more frequent confusion or memory loss, is prevalent among individuals with COPD, with approximately 36% of them experiencing at least mild cognitive impairment [11]. This cognitive impairment not only affects their ability to manage the disease and adhere to medication but also increases risks for depression and anxiety that can further impact self-management and quality of life [12,13]. The absence of self-management can result in a greater risk of hospitalization due to breathing exacerbations, which accelerates a decline in lung function that adversely impacts mobility and, thus, ADLs [14,15]. 

SCD-related functional limitations, defined as the inability to participate in ADLs or social activities, are more significant among people with COPD than those without, with a substantial proportion of people with COPD requiring assistance in various domains due to these limitations [16]. Further, SCD affects individuals with COPD differently based on demographic characteristics such as race, ethnicity, employment status, and health risk behaviors, with disparities observed in the prevalence of SCD among different racial and ethnic groups [12,16,17]. The presence of at least one chronic comorbidity alongside SCD is associated with a heightened risk of SCD-related functional limitations [12]. Individuals with COPD often experience premature diagnosis of comorbidities, which consequently increases their susceptibility to SCD-related functional limitations at an earlier stage in life [12,18]. Older adults with a history of COPD, stroke, and heart disease are most at risk for SCD-risk complications [12], such as difficulties with ADLs and social activities, which are associated with the risk of mortality in this population [19]. Consequently, the burden of managing COPD with functional limitations often falls onto friends, family members, and caregivers, especially among older adults with COPD [20]. 

Given the increasing national burden of COPD [2] and the lack of studies examining the demographic risk factors associated with specific SCD-related functional limitations among people with COPD, it is essential to identify which segments of patients with COPD are most at risk of experiencing these limitations based on demographic and behavioral risk factors [21]. This knowledge is crucial for informing caregivers, health educators, and medical professionals about risk factors and preventative measures, ultimately leading to more targeted interventions for reducing associated functional limitations. The present study analyzed national US Behavioral Risk Factor Surveillance System (BRFSS) data to evaluate the prevalence of SCD-related functional limitations among US adults aged 45 and older with COPD and identify associated sociodemographic risk factors. Further, the study investigated associations between SCD-related functional limitations and health risk behaviors, including smoking and alcohol misuse, among this cohort.

## 2. Materials and Methods

### 2.1. Study Design

In this study, we conducted a secondary analysis of data obtained from the BRFSS in 2019. The BRFSS is a random-digit-dial telephone survey conducted by the Centers for Disease Control and Prevention (CDC) to gather information from noninstitutionalized individuals aged 18 years or older across the United States (US). The survey employs a nationally representative sampling method, with participants being randomly selected to ensure the generalizability of the findings [22,23]. The BRFSS comprises a core component distributed across all US states and optional modules that states can administer based on their specific requirements. In 2019, the Cognitive Decline module was administered with the core component in 25 states. These states include Alabama, Connecticut, the District of Columbia, Florida, Georgia, Indiana, Iowa, Louisiana, Minnesota, Mississippi, Missouri, Nebraska, Nevada, New Mexico, North Dakota, Ohio, Oregon, Pennsylvania, Rhode Island, South Dakota, Tennessee, Texas, Virginia, West Virginia, and Wisconsin. The Cognitive Decline module was exclusively administered to individuals in these states who were aged 45 or older and who answered “yes” to the question, “Have you ever been told that -you have COPD?” and “yes” to the item “During the past 12 months, have you experienced confusion or memory loss that is happening more often or is getting worse?”. The university’s institutional review board deemed the study exempt from patient consent because it involved a secondary analysis of data obtained from BRFSS, which was collected by the CDC and provided in a deidentified format for public use.

### 2.2. Measures

Demographic Variables. The demographic variables included in the analyses were biological sex, age category, race, ethnicity, education level, annual income, and employment status. Biological sex was coded as 0 for females and 1 for males. Age category was categorized as 45–54 years old, 55–64 years old, and 65 years old and above. Race was categorized as White, Black or African American, American Indian/Alaskan Native, Asian, Multiracial, and Other Race. Ethnicity was categorized as Not Hispanic and Hispanic. Education level was categorized as less than high school, high school, and attended college or technical school. Annual income was categorized as USD 0–USD 24,999, USD 25,000–USD 49,999, USD 50,000–USD 74,999, and USD 75,000 and above. Employment status was categorized as employed, unemployed, or retired. 

Health-Related Factors. Health-related factors included self-rated health (SRH) and health risk behaviors. SRH was dichotomized as 0 for fair or poor health and 1 for good or better health. This dichotomization was based on research indicating an association between poor SRH and cognitive decline [24], particularly in older populations [25]. Health-risk behaviors assessed included current smoking (0 = no, 1 = yes) and binge drinking, defined as men having five or more drinks on one occasion in the past 30 days and women having four or more drinks on one occasion in the past 30 days (0 = no, 1 = yes). Assessing binge drinking in studies of US adults with COPD is crucial because over half (50%+) of high-risk patients participating in health-risk behaviors engage in binge drinking, which can intensify respiratory symptoms and contribute to other health complications [26,27]. Additionally, binge drinking often co-occurs with other high-risk behaviors, such as smoking, further amplifying the negative health impacts on cognitive health among older adults [28], including those with COPD [27]. 

Assessment of SCD Impact on ADLs. To understand the impact of SCD on participants’ ADLs, respondents self-reporting SCD were asked specific questions. These questions included inquiries about (a) the frequency of giving up day-to-day household activities or chores, (b) the interference of confusion or memory loss with work, volunteering, or social activities outside the home, and (c) the need for assistance with day-to-day activities due to confusion or memory loss. Respondents who reported “always”, “usually”, or “sometimes” to these questions were coded as responding “yes” to having an SCD-related functional limitation; all other responses were coded as “no”.

### 2.3. Data Analysis

The data analysis employed in this study adhered to the BRFSS survey’s complex sampling design. Appropriate weight, cluster, and stratification variables were incorporated into the model to account for the population-weighted estimates and confidence intervals (CIs) [29,30]. Frequency statistics, presented as percentages with corresponding 95% CIs, were computed for all sociodemographic, health-related behaviors, and SCD-related variables. Logistic regression modeling was then conducted to examine the associations between sociodemographic factors (i.e., sex, age group, race, ethnicity, education level, annual income, employment status), health-related indicators (self-reported health status, binge drinking, current smoking status), and three SCD-related functional limitations reported by individuals with SCD. These functional limitations included needing assistance with day-to-day activities due to memory loss, giving up day-to-day household activities and chores due to memory loss, and confusion or memory loss interfering with social activities, as well as discussing confusion or memory loss with a healthcare professional. 

The data analysis proceeded in two steps to test the relationships between sociodemographic factors, health-related indicators, and SCD-related functional limitations. Firstly, univariable logistic regression was conducted to identify associations between sociodemographic and health-related indicators with each SCD-related functional limitation. Odds ratios, 95% CIs, and *p* values were examined to assess the significance of the predictors and the model fit. The Bonferroni correction was applied, with statistically significant variables (*p* < 0.005) retained for simultaneous entry into the second step of the analysis. In the second step, multivariable logistic regression was conducted to examine the relationships further. The final models were determined based on model fit statistics and odds ratios with 95% CIs. The data analysis was performed using SAS 9.4 for Windows [31], incorporating procedures for complex sampling analysis and weighted data.

## 3. Results

### 3.1. Demographic Characteristics

Data from 107,204 US adults with COPD and SCD were analyzed. Table 1 provides the descriptive statistics for SCD-related items among respondents self-reporting COPD and SCD. Approximately 37% of the respondents (95% CI: 35.5–38.7) had given up day-to-day household activities or chores they used to do. About one-third of the respondents reported that confusion or memory loss interfered with social activities (32%, 95% CI: 30.9–34.1) and necessitated assistance with day-to-day activities (31%, 95% CI: 29.9–32.9%). 

### 3.2. Predictors of Needing Assistance with Day-to-Day Activities Due to Memory Loss

Table 2 presents univariable and multivariable analyses examining the influence of demographics and health-risk behaviors on needing assistance with day-to-day activities due to memory loss.

In the univariable models (see Table 2), Black or African Americans were almost three times more likely than White people to need assistance with daily activities, and Hispanic people were 2.5 times more likely than non-Hispanic people to need assistance. Both unemployed and retired respondents were also more likely than employed respondents to report needing assistance. Current smokers were more likely than non-smokers to need assistance with activities.

On the other hand, males were less likely to need assistance with activities than females and respondents aged 55 to 64 and 65+ were significantly less likely to need assistance with activities than younger respondents between 45 to 54. Those with higher education levels and larger annual incomes were increasingly less likely to report needing assistance with day-to-day activities than respondents who did not graduate from high school and those with annual incomes between USD 0 and USD 24,999 per year, respectively. Respondents in good or better health and those who binge drank were also less likely to report needing assistance with activities than their counterparts. 

In the multivariable model (see Table 2), Asian Americans were more than four times more likely to need assistance than Whites. Respondents who were retired or unemployed were also three to five times more likely to report needing assistance with day-to-day activities due to memory loss than respondents who were employed. Those reporting good or better health and binge drinkers were significantly less likely to report needing assistance than their counterparts. 

### 3.3. Predictors of Giving up Day-to-Day Household Activities and Chores Due to Memory Loss

Table 3 presents univariable and multivariable analyses examining the influence of demographic and health risk behaviors on giving up day-to-day household activities and chores due to memory loss. 

In the univariable analyses (see Table 3), Black or African Americans were almost five times more likely than White people to give up day-to-day chores due to confusion. Hispanic people were three times more likely than non-Hispanic people to give up household chores. Current smokers were more likely than non-smokers to give up household chores. In contrast, males were less likely than females to give up household activities or chores due to memory loss. Those 65 or older were also less likely to give up activities than those 45 to 54. Those with higher education levels and larger annual incomes were increasingly less likely to give up chores than respondents who did not graduate from high school and those with low annual incomes between USD 0 and USD 24,999 per year, respectively. Binge drinkers and those in good or better health were significantly less likely to report giving up chores than their counterparts. 

In the multivariable analysis (see Table 3), Black or African Americans were 4.9 times more likely than White people to give up day-to-day chores due to confusion. In addition, Asian Americans were also 4.6 times more likely than White people to give up day-to-day chores due to memory loss or confusion. Hispanic people were 2.4 times more likely than non-Hispanic people to give up household chores. Unemployed or retired respondents were more likely to report giving up chores than respondents who were employed. Conversely, binge drinkers were significantly less likely to report giving up chores than their counterparts.

### 3.4. Predictors of Confusion or Memory Loss Interfering with Social Activities

Table 4 presents univariable and multivariable analyses examining the associations between demographic and health risk behaviors and confusion or memory loss interfering with social activities. 

In the univariable analyses (see Table 4), Black or African Americans were almost four times more likely to report that confusion or memory loss interfered with their social activities compared to Whites. Hispanics were 1.7 times more likely than non-Hispanics to give up household chores. Unemployed respondents were over five times more likely to report confusion or memory loss interfering with social activities than those employed. 

Current smokers were also more likely than non-smokers to report confusion or memory loss interfering with social activities in the univariable analysis. Compared to adults aged 45 to 54, respondents aged 55 to 64 and those 65 or older were significantly less likely to report that confusion or memory loss interferes with their social activities. Those with higher education levels and larger annual incomes were also less likely to report that confusion or memory loss interferes with their social activities than respondents who did not graduate from high school and those with low annual incomes between USD 0 and USD 24,999 per year, respectively, while those in good or better health and those reporting binge drinking in the past 30 days were less likely to report interference with social activities than those in fair or poor health and non-binge-drinkers, respectively. 

In the multivariable analysis (see Table 4), although the magnitude of the effect was slightly attenuated, Black or African Americans remained over three times more likely to report that confusion or memory loss interfered with their social activities compared to Whites. Those identifying as being of “other race” were also over two times more likely to report that confusion or memory loss interfered with their social activities compared to Whites. Unemployed respondents remained over five times more likely to report confusion or memory loss interfering with social activities than those who were employed, with the magnitude of the effect increasing. Those in good or better health and those reporting binge drinking in the past 30 days continued to be less likely to report confusion or memory loss interfering with social activities than those in fair or poor health and non -binge drinkers, respectively.

## 4. Discussion

This study utilized 2019 BRFSS data to investigate the prevalence of SCD-related functional limitations among older adults with COPD. Our research contributes to the existing literature by uncovering population-based associations between demographic and behavioral risk factors and SCD-related functional limitations in US adults with COPD. Most notably, approximately one-third of the BRFSS respondents in our study reported that confusion or memory loss required them to need assistance with day-to-day activities. This finding is consistent with previous research on cognitive impairment in COPD, which reported that one-third of individuals with COPD and SCD-related limitations sought help from family and friends, and two-thirds required assistance with ADLs [16]. Reliance on assistance from family and friends may lead to frustration about their dependence on others [32]. In the absence of support from others, many individuals with COPD take frequent breaks while dressing or moving to accommodate their functional limitations, ultimately leading to the abandonment of day-to-day activities [32]. Therefore, it is crucial to determine what increases the prevalence of SCD-induced functional limitations and establish care coordination practices to help reduce the burden they place on patients with COPD and their informal caregivers.

Our study also revealed significant racial and ethnic disparities in the prevalence of SCD-related functional limitations among individuals with COPD. Consistent with previous research [33], our findings indicate that Black or African American individuals with COPD are almost five times more likely to report functional difficulty due to cognitive decline and three times more likely to report that memory loss interferes with their social activities compared to their White counterparts. Moreover, our study found that interference with work and social activities is most prevalent in Black and Hispanic people with COPD and SCD and that Black or African American people and Asians with COPD and SCD were more likely to report giving up day-to-day household activities and chores due to cognitive decline than White people. These findings are supported by Chan et al. [34], who also reported higher rates of difficulty with instrumental ADLs in Black adults compared to Latino, non-Hispanic White, and Asian groups. Moreover, our results align with those of Read et al. [35], who showed that problems with memory are associated with older individuals’ withdrawal from social activities. Therefore, strategies are needed to decrease disparities associated with SCD-driven functional limitations in racial minority groups with COPD.

Longitudinal data from the National Health and Examination Survey [36] support our findings that non-Hispanic Black and Latino adults with COPD report higher rates of subjective cognitive concerns than non-Hispanic White people with COPD. This evidence underscores the critical need for targeted interventions and policies to address the specific cognitive challenges faced by Black and Hispanic individuals with COPD. Additionally, our study highlights the necessity of further investigating racial and ethnic disparities related to the impact of SCD to clarify the underlying mechanisms influencing these disparities. Culturally sensitive intervention strategies can be developed for non-Hispanic Black and Latino adults with COPD to help them overcome functional limitations caused by cognitive decline and memory loss. The inclusion of racially and culturally conscious interventions and policies will advance efforts toward diminishing health disparities surrounding cognitive decline for non-Hispanic Black and Latino populations with COPD. 

Results of our study also indicate that individuals with COPD and SCD who were retired or unemployed were significantly more likely to report functional difficulties due to memory loss compared to their employed counterparts. Additionally, unemployed adults with COPD and SCD were also significantly more likely to report memory loss interfering with social activities. Moreover, our study revealed that individuals with COPD and SCD who were unemployed or retired were more likely to report giving up day-to-day household activities and chores than their employed counterparts. These findings are consistent with previous research that has shown the impact of employment status on the well-being and functional abilities of individuals with chronic illnesses. For instance, Abdelwahab et al. [37] found that a significant proportion of patients with COPD gave up their jobs due to their condition, highlighting the influence of employment on the lives of individuals with chronic respiratory diseases. Group-based interventions using pulmonary rehabilitation and regular physical activity are needed for unemployed and retired patients with COPD to explore the extent to which reestablishing or maintaining a social connection can improve SCD-related functional limitations.

Further, Jacobsen et al. [38] demonstrated a relationship between lack of paid work and characteristics such as low daily physical activity and poor nutritional status in patients with COPD, indicating the potential impact of employment on functional abilities. Epidemiological analyses support that about one-quarter of adults with COPD are unable to work [15]. Paid employment is a protective factor for physical [39] and cognitive [40] functioning, while unemployment has been identified as a strong predictor of poor cognitive function, particularly in rural populations with COPD [40]. Addressing the challenges of unemployed or retired individuals with these conditions is crucial for developing comprehensive care strategies that support their functional independence and overall well-being [41,42]. Additional research is warranted to examine how unemployment interacts with cognitive decline to impact functional impairment among individuals with COPD. 

Our study also found that binge drinkers with COPD were less likely to report memory loss interfering with social activities than non-binge drinkers. These results diverge from established associations between COPD, binge drinking, and cognitive decline [43], suggesting the potential underreporting of memory problems in binge drinkers with COPD, possibly due to denial, stigma, or other factors. However, alcohol consumption tends to increase in social settings [44]. Older adults with COPD who engage in binge drinking may still participate in social interactions despite experiencing some memory loss. The complex interplay between health risk behaviors, memory loss, and social activities engaged in by individuals with COPD requires further investigation to untangle the associations between these variables. Once the underlying mechanisms are further understood, more tailored interventions can be developed for older adults with COPD who display drinking patterns that are likely to impact both memory decline and social interactions.

### Limitations

Using the BRFSS data presents several limitations that warrant consideration. The cross-sectional nature of the BRFSS data limits the ability to establish causal relationships between variables from this secondary analysis. The Cognitive Decline module was administered in only 25 states, potentially constraining the generalizability of the findings to the entire US population. The exclusion of individuals under 45 years old from the Cognitive Decline module may also introduce age-related bias, as cognitive decline can also affect younger individuals with COPD. Moreover, the diagnoses of COPD and SCD were self-reported by respondents and not confirmed by their healthcare provider. Reliance on self-reported data may introduce response, misclassification, and social desirability biases, potentially impacting the accuracy of the reported functional limitations associated with cognitive decline. The exclusion of individuals living in care facilities with COPD and SCD due to the BRFSS’s administration exclusively to noninstitutionalized adults may lead to an underrepresentation of individuals with severe cognitive impairment, limiting the comprehensiveness of the study’s findings. Despite these limitations, the statistical analysis plan mitigated the potential for biases. Additionally, the large sample size, statistical analysis that included unadjusted and adjusted logistic regression analyses, and the nationally representative nature of the BRFSS data enhance the generalizability of the study’s results to the broader population of US adults with COPD. Nevertheless, the potential biases and limitations inherent in the BRFSS data should be considered when interpreting the results.

## 5. Conclusions

The noteworthy prevalence of SCD-related functional limitations among adults with COPD highlights the importance of early detection to prevent or manage cognitive impairment-associated functional issues. Most notably, racial and ethnic disparities in the prevalence of SCD-related functional limitations portend the need for more targeted interventions and policies to address challenges faced by Black/African Americans and Hispanics with COPD. Additionally, retired and unemployed individuals with COPD and SCD may have unmet needs related to ADLs due to memory loss. The intricate interplay between health status, memory loss, and social engagement underscores the critical need for healthcare workers to screen patients with COPD for SCD, particularly those belonging to high-risk demographic and behavioral subgroups. 

## Figures and Tables

**Table 1 ijerph-21-00278-t001:** Descriptive statistics for SCD and SCD-related items among respondents ≥45 years old diagnosed with COPD, 2019 Behavioral Risk Factor Surveillance System.

	N	%	95% CI (%)
Subjective Cognitive Decline (SCD)			
Yes	11,847	11	10.8–11.5
No	94,613	89	88.5–89.2
SCD-related functional limitations			
Need assistance with said day-to-day activities due to memory loss or confusion			
Yes	3750	31	29.9–32.9
No	8667	67	67.1–70.1
Given up day-to-day household activities or chores you used to do due to confusion or memory loss			
Yes	4366	37	35.5–38.7
No	7952	63	61.3–64.5
Confusion or memory loss interferes with social activities			
Yes	3815	32	30.9–34.1
No	8472	68	66.0–69.1

**Table 2 ijerph-21-00278-t002:** Univariable and multivariable logistic regression results for adults with COPD who report needing assistance with day-to-day activities due to memory loss.

	Univariable	Multivariable
Predictor	OR (95% CI)	AOR (95% CI)
Sex		
Female ^Ref^	1.0	1.0
Male	0.8 (0.7–0.9) ***	1.3 (0.6–2.5)
Age		
45 to 54 ^Ref^	1.0	1.0
55 to 64	0.8 (0.6–0.95) *	0.5 (0.2–1.2)
65 or older	0.4 (0.4–0.5) ***	0.6 (0.3–1.5)
Race		
White ^Ref^	1.0	1.0
Black or African American	2.9 (1.4–6.1) ***	2.2 (0.9–5.6)
American Indian/Alaskan Native	0.4 (0.1–1.2)	0.5 (0.2–1.4)
Asian	2.1 (0.6–7.3)	4.4 (1.5–12.9) **
Multiracial	0.8 (0.3–2.3)	0.8 (0.2–3.7)
Other race	0.9 (0.4–2.1)	1.0 (0.4–2.6)
Ethnicity		
Not Hispanic ^Ref^	1.0	1.0
Hispanic	2.5 (1.9–3.5) ***	1.3 (0.7–2.5)
Education		
Did not graduate from HS ^Ref^	1.0	1.0
HS Graduate	0.6 (0.4–0.7) ***	0.6 (0.3–1.3)
Attended College or Tech School	0.4 (0.3–0.5) ***	0.7 (0.3–1.4)
Grad from College or Tech School	0.2 (0.2–0.3) ***	0.8 (0.3–2.2)
Income		
USD 0 to USD 24,999 K ^Ref^	1.0	1.0
USD 25 to USD 49,999 K	0.5 (0.4–0.6) ***	0.6 (0.3–1.4)
USD 50 K to 74,999 K	0.2 (0.2–0.3) ***	0.8 (0.4–1.9)
USD 75 K+	0.2 (0.2–0.3) ***	0.5 (0.2–1.2)
Employment		
Employed ^Ref^	1.0	1.0
Unemployed	6.9 (5.4–8.8) ***	5.8 (3.0–11.5) ***
Retired	1.6 (1.3–2.1) ***	3.0 (1.2–8.0) *
Self-Reported Health		
Fair or Poor Health ^Ref^	1.0	1.0
Good or Better Health	0.3 (0.2–0.3) ***	0.5 (0.3–0.99) *
Binge Drinking		
No ^Ref^	1.0	1.0
Yes	0.5 (0.4–0.6) ***	0.6 (0.3–0.97) *
Current Smoker		
No ^Ref^	1.0	1.0
Yes	2.4 (2.0–2.8) ***	1.3 (0.8–2.3)

* *p* < 0.05; ** *p* < 0.01; *** *p* < 0.005; ^Ref^ = Reference values; CI = confidence interval; OR = odds ratio; AOR = adjusted odds ratio.

**Table 3 ijerph-21-00278-t003:** Univariable and multivariable logistic regression results for adults with COPD who report functional difficulty due to cognitive decline.

	Univariable	Multivariable
Predictor	OR (95% CI)	AOR (95% CI)
Sex		
Female ^Ref^	1.0	1.0
Male	0.7 (0.6–0.9) ***	0.9 (0.5–1.5)
Age		
45 to 54 ^Ref^	1.0	1.0
55 to 64	0.8 (0.7–1.0)	0.7 (0.4–1.4)
65 or older	0.4 (0.4–0.5) ***	0.9 (0.5–2.1)
Race		
White ^Ref^	1.0	1.0
Black or African American	4.7 (2.3–9.3) ***	4.9 (2.3–10.4) ***
American Indian/Alaskan Native	2.1 (0.4–10.6)	4.0 (0.3–49.0)
Asian	1.7 (0.5–5.9)	4.6 (1.5–14.3) **
Multiracial	0.9 (0.3–2.3)	1.1 (0.3–3.7)
Other race	1.1 (0.5–2.5)	1.1 (0.4–3.7)
Ethnicity		
Not Hispanic ^Ref^	1.0	1.0
Hispanic	3.0 (2.1–4.1) ***	2.4 (1.2–4.7) *
Education		
Did not graduate from HS ^Ref^	1.0	1.0
HS Graduate	0.6 (0.5–0.7) ***	0.5 (0.2–1.2)
Attended College or Tech School	0.4 (0.3–0.5) ***	0.7 (0.3–1.7)
Grad from College or Tech School	0.2 (0.2–0.3) ***	0.5 (0.2–1.3)
Income		
USD 0 to USD 24,999 K ^Ref^	1.0	1.0
USD 25 K to USD 49,999 K	0.4 (0.3–0.5) ***	0.7 (0.3–1.4)
USD 50 K to 74,999 K	0.2 (0.2–0.3) ***	0.6 (0.2–1.4)
USD 75 K+	0.2 (0.1–0.2) ***	0.6 (0.3–1.4)
Employment		
Employed ^Ref^	1.0	1.0
Unemployed	5.9 (4.8–7.3) ***	7.7 (4.0–14.8) ***
Retired	1.4 (1.1–1.7) ***	3.1 (1.3–7.9) *
Self-Reported Health		
Fair or Poor Health ^Ref^	1.0	1.0
Good or Better Health	0.3 (0.2–0.3) ***	0.8 (0.4–1.5)
Binge Drinking		
No ^Ref^	1.0	1.0
Yes	0.5 (0.4–0.5) ***	0.6 (0.4–0.9) *
Current Smoker		
No ^Ref^	1.0	1.0
Yes	2.3 (1.9–2.7) ***	0.8 (0.5–1.3)

* *p* < 0.05; ** *p* < 0.01; *** *p* < 0.005; ^Ref^ = Reference values; CI = confidence interval; OR = odds ratio; AOR = adjusted odds ratio.

**Table 4 ijerph-21-00278-t004:** Univariable and multivariable logistic regression results for adults with COPD who report that memory loss interferes with social activities.

	Univariable	Multivariable
Predictor	OR (95% CI)	AOR (95% CI)
Sex		
Female ^Ref^	1.0	1.0
Male	0.9 (0.8–1.0)	
Age		
45 to 54 ^Ref^	1.0	1.0
55 to 64	0.7 (0.6–0.9) ***	0.9 (0.5–1.6)
65 or older	0.3 (0.3–0.4) ***	0.6 (0.3–1.4)
Race		
White ^Ref^	1.0	1.0
Black or African American	3.9 (2.2–6.8) ***	3.6 (2.0–6.4) ***
American Indian/Alaskan Native	0.5 (0.1–1.6)	0.7 (0.2–2.0)
Asian	2.0 (0.6–6.9)	2.7 (0.9–7.8)
Multiracial	0.9 (0.3–2.4)	0.9 (0.3–3.2)
Other race	1.2 (0.5–2.8)	2.5 (1.1–6.0) *
Ethnicity		
Not Hispanic ^Ref^	1.0	1.0
Hispanic	1.7 (1.2–2.4) **	0.7 (0.4–1.4)
Education		
Did not graduate from HS ^Ref^	1.0	1.0
HS Graduate	0.6 (0.5–0.7) ***	0.6 (0.3–1.3)
Attended College or Tech School	0.5 (0.4–0.6) ***	0.6 (0.3–1.4)
Grad from College or Tech School	0.3 (0.3–0.4) ***	0.6 (0.2–1.5)
Income		
USD 0 to USD 24,999 K ^Ref^	1.0	1.0
USD 25 to USD 49,999 K	0.5 (0.4–0.6) ***	1.2 (0.6–2.2)
USD 50 K to 74,999 K	0.3 (0.2–0.3) ***	1.4 (0.6–3.3)
USD 75 K+	0.3 (0.2–0.3) ***	1.5 (0.8–3.2)
Employment		
Employed ^Ref^	1.0	1.0
Unemployed	5.0 (4.1–6.2) ***	5.7 (2.9–11.0) ***
Retired	1.0 (0.8–1.3)	2.3 (0.9–5.8)
Self-Reported Health		
Fair or Poor Health ^Ref^	1.0	1.0
Good or Better Health	0.3 (0.2–0.3) ***	0.4 (0.2–0.7) ***
Binge Drinking		
No ^Ref^	1.0	1.0
Yes	0.6 (0.5–0.7) ***	0.6 (0.3–0.9) *
Current Smoker		
No ^Ref^	1.0	1.0
Yes	2.6 (2.2–3.1) ***	0.9 (0.6–1.7)

* *p* < 0.05; ** *p* < 0.01; *** *p* < 0.005; ^Ref^ = Reference values; CI = confidence interval; OR = odds ratio; AOR = adjusted odds ratio.

## Data Availability

Publicly available datasets were analyzed in this study. The BRFSS data analyzed during this study can be found here: https://www.cdc.gov/brfss/annual_data/annual_2019.html (accessed on 26 February 2024).

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
