# Peer review of "Demographic and Behavioral Risk Factors Predict Functional Limitations Associated with Subjective Cognitive Decline in Americans with Chronic Obstructive Pulmonary Disease: A Secondary Analysis"

_ijerph, 2024, doi:10.3390/ijerph21030278_

Round 1

Reviewer 1 Report

Comments and Suggestions for Authors

Thank you for your hard work in analyzing the data from the Behavioral Risk Factor Surveillance System, in particular relating to cognitive function in people with COPD. Cognitive function is a very interesting area and as you have rightly said may influence patients’ ability to participate in health self-management. I have made a few comments below.

Methods:

Study design

·         In your description of the survey, I am wondering if the survey was administered on-line or via a paper survey. Important to clarify this.

·         Was an Ethics application submitted to analyze this data?

Results:

Demographic characteristics

·         Line 174, 176 - 178:Approximately 37% of the respondents (95% CI: 35.46% - 38.71%) had given …’ There is no need to add percentages to CI numbers e.g., (95% CI: 35.5 - 38.7) is more correct.

·         Also decrease the number of decimal points

·         Table 1: for percentage results make whole numbers i.e., no decimals for percentages

·         To make the Table 1 less cluttered and without reducing accuracy, make present percentages as whole numbers, remove % sign from 95%CI e.g.

11,847

11

10.8 - 11.5

Predictors of needing assistance with day-to-day activities due to memory loss

·         When reporting results – you do not need to write the numbers in the paragraph and in Table 2. There are lots of interesting results here, so make comments and refer to the Table for the numbers, as you have done in the previous paragraph.

·         Table 2 heading: make a more succinct heading and any abbreviations need to be in the Key/legend below the Table. Heading example ‘Results for people needing assistance with day-to-day activities due to memory loss’.

·         Table 2: Make 95%CI to one decimal place where possible – sometimes not possible.

Predictors of giving up day-to-day household activities and chores due to memory loss

·         As above – don’t put the numbers in the paragraph and the Table. Describe the results in words and then put numbers in the Table. Example below.

·         Table 3: write a more succinct heading. Same comments as for the above Tables

Predictors of confusion or memory loss interfering with social activities

·         Line 251: Suggestion for writing results without the numbers – as long as all the numbers are in the Table. In the univariate analyses, Black or African Americans were almost four times more likely to report that confusion or memory loss interfered with their social activities compared to whites. (OR = 3.90, 95% CI: 2.24-6.79, P<0.005). Hispanics were 1.69 times more likely than non-Hispanics to give up household chores than non-Hispanics. (95% CI: 1.22-254 2.35, P<0.01). Unemployed respondents were over five times more likely to report confusion or memory loss interfering with social activities than those employed. (OR = 5.04, 95% 256 CI: 4.11-6.20, P<0.005).

·         Table 4: Make the heading more succinct as in other Tables e.g., ‘Results for how memory loss interferes with social activities in adults with COPD’ by demographic and health risk factors with AORs of interference with social activities by age, race, education level, annual income, employment status, health status, binge drinking, and smoking status, BRFSS, 2019.

Thank you again – I have thoroughly enjoyed reading your manuscript especially the results and discussion sections.

Author Response

Comment: Thank you for your hard work in analyzing the data from the Behavioral Risk Factor Surveillance System, in particular relating to cognitive function in people with COPD. Cognitive function is a very interesting area and as you have rightly said may influence patients’ ability to participate in health self-management. I have made a few comments below.

  • Response: Thank you for your support of our work. We appreciate the time, thought, and effort put into your feedback.

Comment: Methods - Study design: In your description of the survey, I am wondering if the survey was administered on-line or via a paper survey. Important to clarify this.

  • Response: BRFSS is conducted via telephone. To address this comment, on page 2 on line 95, the authors added this clarification in the following statement:

“The BRFSS is a random-digit-dial telephone survey conducted by the Centers for Disease Control and Prevention (CDC) to gather information from noninstitutionalized individuals aged 18 years or older across the United States (US).”

Comment: Was an Ethics application submitted to analyze this data?

  • Response: On page 3 on lines 110-113, the authors addressed the Ethics application. Because this is a secondary analysis of deidentified data, the study was exempt from review:

“The university’s institutional review board deemed the study exempted because it involved a secondary analysis of data obtained from BRFSS, which was collected by the CDC and provided in a deidentified format for public use.”

Comment: Results - Demographic characteristics: Line 174, 176 - 178: ‘Approximately 37% of the respondents (95% CI: 35.46% - 38.71%) had given …’ There is no need to add percentages to CI numbers e.g., (95% CI: 35.5 - 38.7) is more correct. Also decrease the number of decimal points

  • Response: We have reduced the number of decimal points to 1 for the 95% CI for the percentages that are reported (i.e., Table 1). The two decimal points next to CI numbers in lines 177, 179-181 have been reduced to one.

Comment: Table 1: for percentage results make whole numbers i.e., no decimals for percentages

  • Response: The percentage results were rounded to the nearest whole number in Table 1 (p. 4-5).

Comment: To make the Table 1 less cluttered and without reducing accuracy, make present percentages as whole numbers, remove % sign from 95%CI e.g.

11,847

11

10.8 - 11.5

  • Response: The percentages were rounded to the nearest whole number and CI percentage signs were removed and reduced to one decimal place in Table 1 (p. 4-5)

Comment: Predictors of needing assistance with day-to-day activities due to memory loss: When reporting results – you do not need to write the numbers in the paragraph and in Table 2. There are lots of interesting results here, so make comments and refer to the Table for the numbers, as you have done in the previous paragraph.

  • Response: The numbers within the paragraphs of lines 190-195 and 197-205 have been removed. The authors added the following statements to line 188 and line 213 respectively to refer the readers to Table 2: “In the univariable models (see Table 2), Black or African Americans were almost three times more likely than Whites to need assistance with daily activities and Hispanics were 2.5 times more likely than non-Hispanics to need assistance,” and “In the multivariable model (see Table 2), Asian Americans were more than four times more likely to need assistance than Whites.”

Comment: Table 2 heading: make a more succinct heading and any abbreviations need to be in the Key/legend below the Table. Heading example ‘Results for people needing assistance with day-to-day activities due to memory loss’.

  •  Response: The heading for Table 2 (lines 210-212) was changed to “Univariable and multivariable logistic regression results for adults with COPD who report needing assistance with day-to-day activities due to memory loss.” Abbreviations for odds ratios and adjusted odds ratios were added to the key/legend below Table 2 (p. 5-6).

Comment: Table 2: Make 95% CI to one decimal place where possible – sometimes not possible.

  • Response: The CIs in Table 2 have been changed to one decimal point where possible (p. 5-6).

Comment: Predictors of giving up day-to-day household activities and chores due to memory loss: As above – don’t put the numbers in the paragraph and the Table. Describe the results in words and then put numbers in the Table. Example below.

  • Response: The numbers within the paragraphs of lines 226-238 and 246-253 have been removed. The authors added the following statements to line 225 and line 245 respectively to refer the readers to Table 3: “In the univariable analyses (see Table 3), Black or African Americans were almost five times more likely than Whites to give up day-to-day chores due to confusion,” and “In the multivariable analysis (see Table 3), Black or African Americans were 4.9 times more likely than Whites to give up day-to-day chores due to confusion.”

Comment: Table 3: write a more succinct heading. Same comments as for the above Tables

  • Response: The heading for Table 3 (lines 242-244) was changed to “Univariable and multivariable logistic regression results for adults with COPD who report functional difficulty due to cognitive decline.” Abbreviations for odds ratios and adjusted odds ratios were added to the key/legend below Table 3 (p. 8).

Comment: Predictors of confusion or memory loss interfering with social activities: Line 251: Suggestion for writing results without the numbers – as long as all the numbers are in the Table. ‘In the univariate analyses, Black or African Americans were almost four times more likely to report that confusion or memory loss interfered with their social activities compared to whites. (OR = 3.90, 95% CI: 2.24-6.79, P<0.005). Hispanics were 1.69 times more likely than non-Hispanics to give up household chores than non-Hispanics. (95% CI: 1.22-254 2.35, P<0.01). Unemployed respondents were over five times more likely to report confusion or memory loss interfering with social activities than those employed. (OR = 5.04, 95% 256 CI: 4.11-6.20, P<0.005).

  • Response: The numbers within the paragraphs of lines 261-277 and 287-294 have been removed. The authors added the following statements to line 259 and line 284 respectively to refer the readers to Table 4: “In the univariable analyses (see Table 4), Black or African Americans were almost four times more likely to report that confusion or memory loss interfered with their social activities compared to Whites,” and “In the multivariable analysis (see Table 4), although the magnitude of the effect was slightly attenuated, Black or African Americans remained over three times more likely to report that confusion or memory loss interfered with their social activities compared to Whites.”

Comment: Table 4: Make the heading more succinct as in other Tables e.g., ‘Results for how memory loss interferes with social activities in adults with COPD’ by demographic and health risk factors with AORs of interference with social activities by age, race, education level, annual income, employment status, health status, binge drinking, and smoking status, BRFSS, 2019.

  •  Response: The heading for Table 4 (lines 281-283) was changed to “Univariable and multivariable logistic regression results for adults with COPD who report that memory loss interferes with social activities.” Abbreviations for odds ratios and adjusted odds ratios were added to the key/legend below Table 4 (p. 9-10).

Comment: Thank you again – I have thoroughly enjoyed reading your manuscript especially the results and discussion sections.

  • Response: Thank you!!! Your feedback has greatly improved the quality of our manuscript. We sincerely appreciate all of the time spent helping to review our work.

Reviewer 2 Report

Comments and Suggestions for Authors

Title of the article: Demographic and behavioral risk factors predict functional limitations associated with subjective cognitive decline in Americans with Chronic Obstructive Pulmonary Disease: A secondary analysis

My comments

                In general, this is an interesting research. The authors aim to identify the association between sociodemographic risk factors and SCD-related functional limitations. However, there are major points should be addressed.    

1. Introduction

            Clear and well written.

2. Materials and Methods

                You state that the responses of the assessment of SCD Impact on ADLs were classified as “always,” “usually,” or “sometimes”. Why the response of “yes” or “no” were reported in the results section.

                Please change “univariate” to “univariable” and “multivariate” to “multivariable” as a whole of manuscript.

3. Results

                Please re-write the results section. There are many duplicate the results section in text and table.

4. Discussion

            The diagnosis of COPD and SCD were diagnosed by the questionnaires which was not confirm by doctors. Thus, this should be mention as a limitation of this study.

5. Conclusion

                The conclusion section is too long. Please summarize.  

Comments on the Quality of English Language

Title of the article: Demographic and behavioral risk factors predict functional limitations associated with subjective cognitive decline in Americans with Chronic Obstructive Pulmonary Disease: A secondary analysis

My comments

                In general, this is an interesting research. The authors aim to identify the association between sociodemographic risk factors and SCD-related functional limitations. However, there are major points should be addressed.    

1. Introduction

            Clear and well written.

2. Materials and Methods

                You state that the responses of the assessment of SCD Impact on ADLs were classified as “always,” “usually,” or “sometimes”. Why the response of “yes” or “no” were reported in the results section.

                Please change “univariate” to “univariable” and “multivariate” to “multivariable” as a whole of manuscript.

3. Results

                Please re-write the results section. There are many duplicate the results section in text and table.

4. Discussion

            The diagnosis of COPD and SCD were diagnosed by the questionnaires which was not confirm by doctors. Thus, this should be mention as a limitation of this study.

5. Conclusion

                The conclusion section is too long. Please summarize.  

Author Response

Comment: In general, this is an interesting research. The authors aim to identify the association between sociodemographic risk factors and SCD-related functional limitations. However, there are major points should be addressed.    

  • Response: Thank you for your support of our research. Below you will find our responses to your insightful feedback.

Comment: 1. Introduction - Clear and well written.

  • Response: Thank you very much for your feedback.

Comment: 2. Materials and Methods -  You state that the responses of the assessment of SCD Impact on ADLs were classified as “always,” “usually,” or “sometimes”. Why the response of “yes” or “no” were reported in the results section.

  • Response: Thank you for this assessment. The authors added a clarifying statement to the sentence on page 3 lines 143-145 to address this issue by saying the following: “Respondents who reported “always,” “usually,” or “sometimes” to these questions were coded as “yes” for having an SCD-related functional limitation; all other responses were coded as “no.”

Comment: Please change “univariate” to “univariable” and “multivariate” to “multivariable” as a whole of manuscript.

  • Response: “Univariate” was changed to “univariable” in lines 163, 185, 188, 222, 225, 256, 259, and 267 and in Table 2, Table 3, and Table 4. “Multivariate” was changed to “multivariable” in lines 168, 185, 213, 222, 245, 256, and 284 and in Table 2, Table 3, and Table 4.

Comment: 3. Results - Please re-write the results section. There are many duplicate the results section in text and table.

  • Response: The duplicates from the table and text have been removed from each section.
    • The numbers within the paragraphs of lines 177-181 and 190-205 have been removed. The authors added the following statements to line 188 and line 213 respectively to refer the readers to Table 2: “In the univariable models (see Table 2), Black or African Americans were almost three times more likely than Whites to need assistance with daily activities and Hispanics were 2.5 times more likely than non-Hispanics to need assistance,” and “In the multivariable model (see Table 2), Asian Americans were more than four times more likely to need assistance than Whites.”
    • The numbers within the paragraphs of lines 226-238 and 246-253 have been removed. The authors added the following statements to line225 and line 245 respectively to refer the readers to Table 3: “In the univariable analyses (see Table 3), Black or African Americans were five times more likely than Whites to give up day-to-day chores due to confusion,” and “In the multivariable analysis (see Table 3), Black or African Americans were 4.9 times more likely than Whites to give up day-to-day chores due to confusion.”
    • The numbers within the paragraphs of lines 261-277 and 287-294 have been removed. The authors added the following statements to line 259 and line 284 respectively to refer the readers to Table 4: “In the univariable analyses (see Table 4), Black or African Americans were almost four times more likely to report that confusion or memory loss interfered with their social activities compared to Whites,” and “In the multivariable analysis (see Table 4), although the magnitude of the effect was slightly attenuated, Black or African Americans remained over three times more likely to report that confusion or memory loss interfered with their social activities compared to Whites.”

Comment: 4. Discussion - The diagnosis of COPD and SCD were diagnosed by the questionnaires which was not confirm by doctors. Thus, this should be mention as a limitation of this study.

  • Response: To address this limitation, the authors added a clarifying statement to the sentence in lines 389-393: “Moreover, the diagnoses of COPD and SCD were self-reported by respondents and not confirmed by their healthcare provider. Reliance on self-reported data may introduce response, misclassification, and social desirability biases, potentially impacting the accuracy of the reported functional limitations associated with cognitive decline.”

Comment: 5. Conclusion - The conclusion section is too long. Please summarize.  

  • Response: The following has been added as a summarized version of the conclusion paragraph (lines 406-415): “The noteworthy prevalence of SCD-related functional limitations among adults with COPD highlights the importance of early detection to prevent or manage cognitive impairment-associated functional issues. Most notably, racial and ethnic disparities in the prevalence of SCD-related functional limitations portend the need for more targeted interventions and policies to address challenges faced by Black/African Americans and Hispanics with COPD. Additionally, retired and unemployed individuals with COPD and SCD may have unmet needs related to ADLs due to memory loss. The intricate interplay between health status, memory loss, and social engagement underscores the critical need for healthcare workers to screen patients with COPD for SCD, particularly those belonging to high-risk demographic and behavioral subgroups.”

Round 2

Reviewer 2 Report

Comments and Suggestions for Authors

All of my comments have been addressed by authors. This manuscript can be accepted for publication.